# STRUCTURE-AWARE TRANSFORMER POLICY FOR INHOMOGENEOUS MULTI-TASK REINFORCEMENT LEARNING

**Sunghoon Hong**[1,3]**, Deunsol Yoon**[1,3]**, Kee-Eung Kim**[1,2]
[1]Kim Jaechul Graduate School of AI, KAIST, Daejeon, Republic of Korea
[2]School of Computing, KAIST, Daejeon, Republic of Korea
[3]LG AI Research, Seoul, Republic of Korea
{shhong01, solpino, kekim}@kaist.ac.kr

## ABSTRACT

Modular Reinforcement Learning, where the agent is assumed to be morphologically structured as a graph, for example composed of limbs and joints, aims to learn a policy that is transferable to a structurally similar but different agent. Compared to traditional Multi-Task Reinforcement Learning, this promising approach allows us to cope with inhomogeneous tasks where the state and action space dimensions differ across tasks. Graph Neural Networks are a natural model for representing the pertinent policies, but a recent work has shown that their multi-hop message passing mechanism is not ideal for conveying important information to other modules and thus a transformer model without morphological information was proposed. In this work, we argue that the morphological information is still very useful and propose a transformer policy model that effectively encodes such information. Specifically, we encode the morphological information in terms of the traversal-based positional embedding and the graph-based relational embedding. We empirically show that the morphological information is crucial for modular reinforcement learning, substantially outperforming prior state-of-the-art methods on multi-task learning as well as transfer learning settings with different state and action space dimensions.

## 1 INTRODUCTION

Deep reinforcement learning (RL) has made remarkable successes over the last several years, achieving human-level performance in various tasks (Silver et al., 2016; Mnih et al., 2015). However, these are limited by individually training policies for each task, often requiring a large amount of interaction data. Multi-task learning, an approach to training a model jointly from diverse tasks in order to improve learning efficiency and prediction accuracy of task-specific models by exploiting commonalities and differences across tasks, has become prevalent in computer vision (CV) (Dosovitskiy et al., 2021) and natural language processing (NLP) (Devlin et al., 2019; Radford et al., 2019).

In this regard, Multi-Task Reinforcement Learning (MTRL) is a promising approach, but training a policy for multiple tasks in the traditional manner is not straightforward in many cases, e.g., a policy designed for a specific robot cannot be reused for another one with a different embodiment. Therefore, most MTRL methods assume same state and action dimensions across tasks (Rusu et al., 2016; Parisotto et al., 2016; Pinto & Gupta, 2017; Yang et al., 2020; Kalashnikov et al., 2021) and define each task by its own reward function, e.g., grab a cup or move it with a robot arm. Instead, we are interested in a more general setting where the tasks are *inhomogeneous*, i.e., of different state and action space dimensions, also known as incompatible control (Kurin et al., 2021).

One of the popular approaches to inhomogeneous MTRL is to assume a graph structure for the agent, depicted in Figure 1 as an example, where limbs and joints are represented as nodes and edges. In this *modular* setting, Graph Neural Networks (GNNs) provide a natural choice for the model of the policy, since (1) they can process graph-structured input of arbitrary sizes and connections, which allows us to obtain a single policy model that can control any agent with various morphology leading

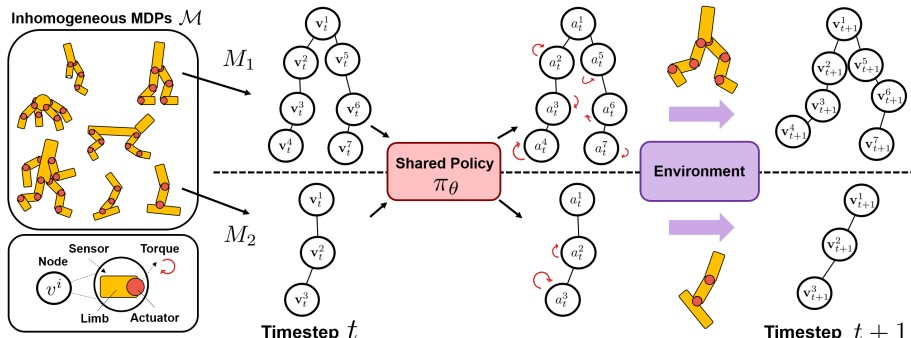

Figure 1: Illustration of modular robot locomotion task. Each node is corresponded to a limb, and each non-torso limb has an actuator to execute an action, which controls the torque of the joint connecting two limbs.

to different state and action space dimensions, and (2) they natively exploit the inductive bias that arises from the morphological structure through their message-passing (MP) scheme, i.e., messages are allowed to be delivered only to nodes connected by edges. Hence, GNN-based approaches have shown to yield significantly better results in inhomogeneous MTRL, compared to the vanilla approaches that don't take into account the morphological structure of the agent.

Recently, Kurin et al. (2021) posed a concern that the MP scheme in GNN is prone to over-smoothing (Li et al., 2018), where the crucial information is washed out during the multi-hop communication. They argued that the advantage of leveraging morphological information for inhomogeneous MTRL was overshadowed by this problem. They therefore advocated adopting the self-attention mechanism, which allows direct communications among nodes, while sacrificing the morphological information. Yet, it is well known that injecting the structural inductive bias into the self-attention mechanism, i.e., positional embedding, can help significantly improve the performance (Vaswani et al., 2017). In the same context, the states of neighbor nodes may be more important than those of non-neighbor nodes for determining actions in modular RL.

In this paper, we introduce Structure-aWAre Transformer (SWAT), a new modular method for inhomogeneous MTRL that effectively incorporates the agent morphology into the transformer-based policy. We propose two forms of structural embeddings of the morphology to be incorporated into the transformer model: (1) traversal-based positional embedding (PE) which represents absolute positions of nodes via tree-traversal algorithms, and (2) graph-based relational embedding (RE) which represents relative distances of node pairs reflecting their connectivity. These allow direct communication of messages among nodes while taking into account the agent morphology.

The experimental results on benchmark tasks for inhomogeneous MTRL strongly support the effectiveness of our structural embedding approach, outperforming prior state-of-the-art methods in terms of sample efficiency and final performance. Through transfer learning experiments, we also demonstrate that the structural information conveyed by our approach plays an important role for downstream tasks.

## 2 BACKGROUND

### 2.1 INHOMOGENEOUS MTRL

We assume a Markov Decision Process (MDP) defined by tuple $M = (\mathcal{S}, \mathcal{A}, \mathcal{T}, \rho_0, r, \gamma)$ to represent an RL task, where $\mathcal{S}$ is the state space, $\mathcal{A}$ is the action space, $\mathcal{T}(s_{t+1}|s_t, a_t)$ is the state transition probability, $\rho_0$ is the initial state distribution, $r_t = r(s_t, a_t) \in \mathbb{R}$ is the reward associated with state and action, and $\gamma \in (0, 1)$ is the discount factor. The objective of RL is to learn a probability distribution over actions conditioned on states, i.e., policy $\pi(a_t|s_t)$, which maximizes the expected discounted return $\mathcal{R} = \mathbb{E}[\sum_{t \geq 0} \gamma^t r_t]$.

In the inhomogeneous MTRL, the agent is tasked with a set of inhomogeneous MDPs $\mathcal{M} = \{M_1, M_2, \cdots, M_K\}$. We say MDPs are *inhomogeneous* when any pair of MDPs have different dimensionalities in their state or action spaces, i.e., $dim(\mathcal{S}_i) \neq dim(\mathcal{S}_j)$ or $dim(\mathcal{A}_i) \neq dim(\mathcal{A}_j)$, where $M_i, M_j \in \mathcal{M}$ and $\mathcal{S}_i, \mathcal{A}_i$ is the state and action space of $M_i$. In the context of MTRL, the goal is to find a policy that maximizes the average expected discounted return over all the environments, $\frac{1}{K} \sum_{i=1}^{K} \mathbb{E}[\mathcal{R}_i]$ where $\mathcal{R}_i$ denotes the expected discounted return in MDP $M_i$.

We now describe modular RL, where the agent can be represented as an undirected graph $\mathcal{G} = (\mathcal{V}, \mathcal{E})$, e.g., the robotic agent with $n$ limbs and $l$ joints as in Figure 1. Each node (vertices) $v^i \in \mathcal{V}$ for $i \in 1...n$ represents a limb, and an undirected edge $e^{i,j} \in \mathcal{E}$ represents connectivity by a joint between a $v^i$ and $v^j$. For the ease of notation, we assume an ordered set of nodes, i.e., a node $v^i$ is $i$-th element in the ordered set. $\mathcal{E}$ can be represented as an adjacency matrix $\mathbf{A} \in \{0, 1\}^{n \times n}$ that describes the connectivity among nodes, i.e., if $v^i, v^j$ are connected, $\mathbf{A}^{i,j} = 1$ and $\sum_{i,j} \mathbf{A}^{i,j} = 2l$. Thus at each time step $t$, the agent observes the state $s_t$, which consists of local sensory information of the limb such as limb type, coordinates, and angular velocity. Denoting the local sensory information of the limb $v^i$ at time step $t$ as vector $\mathbf{v}_t^i$, the state $s_t$ can be represented as $\{\mathbf{v}_t^1, \mathbf{v}_t^2, \ldots, \mathbf{v}_t^n\}$ with morphology $\mathcal{E}$. Given $s_t$, the policy outputs the action $a_t = \{a_t^1, a_t^2, \ldots, a_t^n\}$ where $a_t^i$ is the torque value for the corresponding actuator controlled by the limb $v^i$.

## 2.2 POLICY MODELS FOR INHOMOGENEOUS MTRL

Traditional policy model for RL, e.g., multi-layer perceptron (MLP) policy, is not suitable for inhomogeneous MTRL since it can only handle the state and action spaces with the same dimension. In this regard, GNNs (Hamilton et al., 2017; Wu et al., 2020) have been a natural choice for modeling policies (Wang et al., 2018; Huang et al., 2020), since they can scale to different state and action dimensions as well as incorporate the agent morphology in their message-passing (MP) scheme.

Let us assume a graph $\mathcal{G} = (\mathcal{V}, \mathcal{E})$ where each node $v^i \in \mathcal{V}$ has the corresponding information, i.e., node representation or message $\mathbf{v}^i$. The messages in GNNs are delivered as follows:

$$
\begin{aligned}
&\text{1. MESSAGE AGGREGATION: } \mathbf{m}^i \leftarrow \sigma(\{\mathbf{v}^j : j \in \mathcal{N}_i\}), \quad \forall v^i \in \mathcal{V} \\
&\text{2. NODE UPDATE: } \mathbf{v}_{\text{update}}^i \leftarrow f_\theta(\mathbf{v}^i, \mathbf{m}^i), \quad \forall v^i \in \mathcal{V},
\end{aligned}
\tag{1}
$$

where $\mathcal{N}_i$ is the indices of the neighborhood of $v^i$, $\sigma$ is a message aggregation function, e.g., average or concatenation, $f_\theta$ is a parameterized node update function that computes a new message given the original message $\mathbf{v}^i$ and aggregated message $\mathbf{m}^i$. GNNs consist of the stack of multiple GNN-layers for multi-hop communication, where a single layer operates a single iteration. GNNs are suitable for inhomogeneous MTRL with various agent morphologies, since each node is processed in a modular manner, allowing a single policy to be learned for diverse agents. In particular, Shared Modular Policies (SMP) (Huang et al., 2020) assumes a tree morphology and adopts both-way MP scheme where messages are delivered back and forth between a root and leaves, showing significantly better performance than a standard monolithic policy in inhomogeneous MTRL.

The self-attention mechanism, introduced in the transformer (Vaswani et al., 2017), is another way of passing messages. Unlike GNNs, it assumes that all nodes in a graph are connected, even enabling direct communication among distant nodes. Specifically, each node feature $\mathbf{v}^i \in \mathbb{R}^{d_v}$ is projected to three vectors, $\mathbf{q}^i, \mathbf{k}^i \in \mathbb{R}^{d_k}, \tilde{\mathbf{v}}^i \in \mathbb{R}^{d_v}$, where $\mathbf{q}^i = \mathbf{v}^i \mathbf{W}_\mathbf{q}$, $\mathbf{k}^i = \mathbf{v}^i \mathbf{W}_\mathbf{k}$, $\tilde{\mathbf{v}}^i = \mathbf{v}^i \mathbf{W}_\mathbf{v}$ are a query, key, value vectors respectively, and $\mathbf{W}_\mathbf{q}, \mathbf{W}_\mathbf{k} \in \mathbb{R}^{d_v \times d_k}, \mathbf{W}_\mathbf{v} \in \mathbb{R}^{d_v \times d_v}$ are learnable projection mappings. Messages are then delivered in the self-attention mechanism as:

$$
\alpha^{i,j} = \frac{\mathbf{q}^i \mathbf{k}^{jT}}{\sqrt{d_k}}, \qquad w^{i,j} = \frac{\exp(\alpha^{i,j})}{\sum_{j'} \exp(\alpha^{i,j'})}, \qquad \mathbf{v}_{\text{update}}^i \leftarrow \sum_j w^{i,j} \tilde{\mathbf{v}}^j, \tag{2}
$$

where $\alpha^{i,j}$ is the unnormalized attention score, and $w^{i,j}$ is the attention weight.

When the agent morphology is a sparse graph, which is quite usual for many agents in nature, the crucial information tends to be washed out during multi-hop communication, an undesirable phenomenon known as over-smoothing. In order to avoid the issue, AMORPHEUS (Kurin et al., 2021) adopts the self-attention mechanism and computes the attention weight over all nodes in the graph, yielding better results than the previous GNN-based approaches. However, it discards the morphological information although it is potentially very useful.

## 2.3 POSITIONAL INFORMATION IN TRANSFORMERS

In the transformer, injecting the positional information is essential since the self-attention mechanism is permutation invariant (Vaswani et al., 2017). In the case of NLP, a permutation of tokens can change the whole context and meaning of a sentence. Thus, the structural bias, representing the positions of tokens, is explicitly given as a form of positional embedding (PE), which is found to be significantly useful at learning the contextual representations of words in different positions (Ke et al., 2021; Wang & Chen, 2020). To be specific, the absolute positional information for the $i$-th token $v_i$ in the sequence is given by the absolute PE vector $\mathbf{p}^i$ and it is added to the corresponding token embedding $\mathbf{v}^i$. On the other hand, the relative positional information between two tokens, e.g., relative distance $(i - j)$, is given by relational embedding (RE), $\mathbf{r}^{i,j} \in \mathbb{R}$, which is added to the attention score as $\alpha^{i,j} = \mathbf{q}^i \mathbf{k}^{jT}/\sqrt{d_k} + \mathbf{r}^{i,j}$ (Raffel et al., 2020; Shaw et al., 2018).

## 3 MOTIVATION FOR STRUCTURAL BIAS

AMORPHEUS (Kurin et al., 2021) proposes a plain transformer model without morphological information due to the aforementioned over-smoothing problem. However, we can still incorporate the morphological information into the transformer, i.e., the structural embedding widely utilized in NLP to add the structural bias.

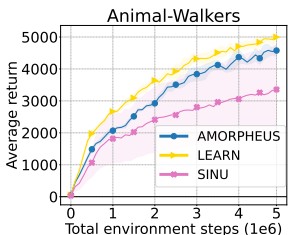

Figure 2: Training curve on `Animal-Walkers`

In order to appreciate its effect in MTRL, we apply 2 common methods used in NLP models, the learnable embedding (**LEARN**) and the sinusoidal encoding (**SINU**). LEARN uses a learnable vector to represent the absolute position of each token in a sequence, while SINU uses a sinusoidal function that can represent the relative position between tokens as well. First, we converted the morphology into a sequence by labeling the index of each node along with the pre-order traversal of the tree with a torso labeled as a root. We then aggregated the sinusoidal encoding or the learnable embedding with the node representations to inject the structural bias into AMORPHEUS. We run AMORPHEUS, LEARN, and SINU on `Animal-Walkers`, which are trained jointly for 5 morphologically different agents.

The results in Figure 2 show how each of the embeddings affects the MTRL performance. LEARN improves the performance the most, because it can roughly capture the knowledge about where each limb is placed. On the other hand, SINU suffers learning instability and the performance rather worsens. The reason for the deterioration, we conjecture, is that SINU encodes morphology incorrectly. SINU, devised to capture a relative distance in sequential data, can give misguided relational information when it is applied to graph-structured data. For example, the distances from a torso to both arms are the same in a graph, but SINU regards them differently. Although this is a preliminary observation, it manifests the necessity of a pertinent embedding scheme for representing both the positional and the relational information of morphology for inhomogeneous MTRL.

## 4 METHOD

Motivated by the result in the previous section, we present the structural embedding scheme that can be effectively incorporated into the transformer-based policy. It allows our policy to leverage the morphological information while free from the over-smoothing. We introduce two components for injecting the structural bias into the transformer: traversal-based PE and graph-based RE.

### 4.1 TRAVERSAL-BASED POSITIONAL EMBEDDING

Unlike language sequences in NLP, it is not straightforward to capture the positional information of each node in a graph. Yet, if we assume tree-structured robot morphology defining a torso as the root node, as done in SMP (Huang et al., 2020), we can represent it as a combination of multiple sequences in terms of tree traversals. We can traverse any tree by several consistent orderings, e.g., pre-order or post-order traversal. Although a single traversal sequence, as in section 3, is ambiguous

to reconstruct the given tree, we can uniquely identify a binary tree with in-order traversal along with another traversal (Burgdorff et al., 1987).

Inspired by this property, we introduce traversal-based PE for representing the position of each node in the given tree. Since we assume an agent morphology as a general tree with an arbitrary number of children, we apply left-child-right-sibling representation (LCRS) to represent a general tree as a binary tree. Then, we traverse the binarized tree by pre-order, in-order, and post-order, which is sufficient to reconstruct the original tree. A node position can be represented as a tuple that consists of the position in each traversal. Concretely, $p_{pre}^{v^i}, p_{in}^{v^i}, p_{post}^{v^i} \in \mathbb{N}$ denote the indices of a node $v^i$ in each traversal ordering respectively, and we assign the PE $\mathbf{p}^i \in \mathbb{R}^{d_v}$ to the node $v^i$ using the learnable embedding vectors $\mathbf{p}_{pre}^{v^i}, \mathbf{p}_{in}^{v^i}, \mathbf{p}_{post}^{v^i}$ as follows:

$$\mathbf{p}^i = \text{AGGREGATE}([\mathbf{p}_{pre}^{v^i}, \mathbf{p}_{in}^{v^i}, \mathbf{p}_{post}^{v^i}]),$$
$$\mathbf{v}^i \leftarrow \mathbf{v}^i + \mathbf{p}^i \tag{3}$$

We simply concatenate the embedding vectors in this work in AGGREGATE. By combining multiple traversals, the PE assigned to each node can uniquely represent the node position in the whole morphology.

## 4.2 GRAPH-BASED RELATIONAL EMBEDDING

In a graph, the relation among nodes also provides structural information from another perspective than the position. Compared to the node positional information above, the relational information contains more explicit knowledge between a pair of nodes. We introduce graph-based RE by utilizing three meaningful features extracted from the graph, which are incorporated into the transformer.

**Normalized Graph Laplacian**  The Graph Laplacian (Merris, 1994) is a matrix that represents connectivity in terms of both adjacency and node degrees in a graph, defined as $\mathbf{L} = \mathbf{D} - \mathbf{A}$ where $\mathbf{A}$ is an adjacency matrix and $\mathbf{D}$ is a diagonal degree matrix, i.e., $\mathbf{D}^{i,i} = |\mathcal{N}_i|$. We use the normalized graph Laplacian to bound its norm, which is defined as $\mathbf{R}_{lap} = \mathbf{D}^{-\frac{1}{2}} \mathbf{L} \mathbf{D}^{-\frac{1}{2}}$. Then, $\mathbf{R}_{lap} \in \mathbb{R}^{n \times n}$ represents the relation among neighbors, capturing local information.

**Shortest Path Distance**  Shortest path distance (SPD) measures the distance between two nodes in a graph, which can be easily computed by several algorithms such as breadth-first search (BFS). It is similar to the relative positional encoding approach in NLP domains, but we use real-value distance $\mathbf{R}_{spd} \in \mathbb{R}^{n \times n}$ divided by the number of nodes $n$ to bound in $[0, 1]$, i.e., $\mathbf{R}_{spd}^{i,j} = SPD(i,j)/n$ where $SPD(i,j)$ denotes the shortest path distance between $v^i, v^j$. SPD considers the direct path among all nodes, so we can say it captures the global information.

**Personalized PageRank**  Personalized PageRank (PPR), a variation of PageRank (Page et al., 1999), represents the proximity between two nodes in a graph based on a random walk model (Park et al., 2019), which is widely utilized in the graph domain (Klicpera et al., 2019; Bojchevski et al., 2020). PPR is a node visitation probability distribution of a $\epsilon$-discounted random walk model with a restarting node. To be concrete, the random walker travels across the Markov chain formed via the graph with its transition probability matrix $\mathbf{P} \in \mathbb{R}^{n \times n}$, where $\mathbf{P}^{i,j} = 1/|\mathcal{N}_i|$ for $\forall v^j \in \mathcal{N}_i$ otherwise 0. The random walker warps to the restarting node with probability $\epsilon$ and follows the $\mathbf{P}$ with $1 - \epsilon$. Then, the node visitation probability distribution conditioned on the restarting node $v^i$, $PPR(i)$, can be obtained by solving a recursive form:

$$\mathbf{x} \leftarrow (1 - \epsilon)\mathbf{x}\mathbf{P} + \epsilon \mathbf{1}_i,$$

$$PPR(i) = \epsilon \mathbf{1}_i (\mathbf{I} - (1 - \epsilon)\mathbf{P})^{-1},$$

where $\mathbf{x}^j$ denotes the probability that the walker resides on $v^j$, and $\mathbf{1}_j$ denotes the one-hot encoded restarting node $v^i$. Then, the proximity between $v^i$ and $v^j$, $\mathbf{R}_{ppr}^{i,j}$, can be denoted by the $j$-th entry of $PPR(i)$. In contrast to SPD, PPR considers every possible path between two nodes, whereby it captures the relation among nodes, without the overall graph dispensed with.

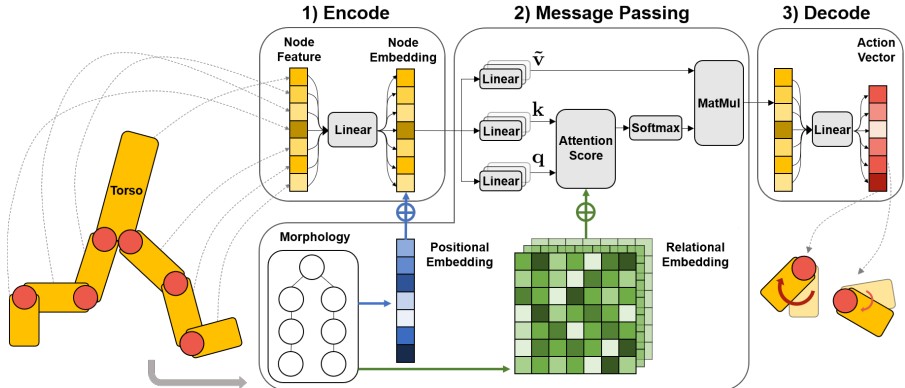

Figure 3: The overview of SWAT.

Although the aforementioned graph features all reflect the agent morphology, they are mutually complementary; for instance, while Graph Laplacian mainly focuses on neighbor information, other features provide knowledge about distant nodes, generating local and global views respectively. We thus represent the relational information among nodes in various perspectives by combining those features. In detail, we aggregate them and learn the graph-based RE $\mathbf{R} \in \mathbb{R}^{n \times n \times n_{head}}$ from them:

$$\mathbf{R}^{i,j} = g_\phi([\mathbf{R}_{lap}^{i,j}, \mathbf{R}_{spd}^{i,j}, \mathbf{R}_{ppr}^{i,j}]), \tag{4}$$

where $g_\phi$ is a function parameterized by $\phi$ that maps the graph features to a RE vector $\mathbf{R}^{i,j}$ and $n_{head}$ denotes the number of heads. Note that we represent the relational information between two nodes as a vector $\mathbf{R}^{i,j} \in \mathbb{R}^{n_{head}}$ instead of a scalar to take full advantage of multi-head attention. Now we modify the attention score formula with the RE in multi-head setting as follows:

$$\alpha_{(h)}^{i,j} = \frac{\mathbf{q}_{(h)}^i \mathbf{k}_{(h)}^{jT}}{\sqrt{d_k/n_{head}}} + \mathbf{R}_{(h)}^{i,j}, \tag{5}$$

where the subscript $(h)$ denotes the $h$-th head and $\mathbf{R}_{(h)}^{i,j}$ is a $h$-th entry of $\mathbf{R}^{i,j}$.

### 4.3 STRUCTURE-AWARE TRANSFORMER POLICY

We now present how SWAT works in inhomogeneous control tasks, illustrated in Figure 3. SWAT is based on a 3-stage framework similar to the existing methods: 1) Encode local sensory information $\mathbf{v}^i$ for each node with the traversal-based PE into the hidden representation by a single MLP layer shared across all nodes. We use the same notation $\mathbf{v}^i$ for the encoded node representation for clear notation. 2) Update each node representation by MP. While SMP uses the multi-hop communication MP scheme, we use the direct communication through self-attention with graph-based RE. 3) Finally, decide the action for each non-torso node by pooling the final node representations. To learn the policy via an RL algorithm, we use TD3 (Fujimoto et al., 2018), a deterministic policy gradient algorithm in the actor-critic framework, following both SMP and AMORPHEUS.

## 5 EXPERIMENT

### 5.1 EXPERIMENT SETTING

We run experiments on modular MTRL benchmarks (Huang et al., 2020; Wang et al., 2018), which are created based on Gym MuJoCo locomotion tasks. 9 environments can be categorized into 2 settings, in-domain and cross-domain. For *in-domain*, there are 4 environments: (1) `Hopper++`, (2) `Walker++`, (3) `Cheetah++`, and (4) `Humanoid++`; they all contain both the intact morphology and its variants, e.g., Humanoid only with one leg or arm, for `Humanoid++`. The *cross-domain* environments are combinations of in-domain environments: (1) `Walker-Humanoid++`, (2) `Walker-Humanoid-Hopper++`,

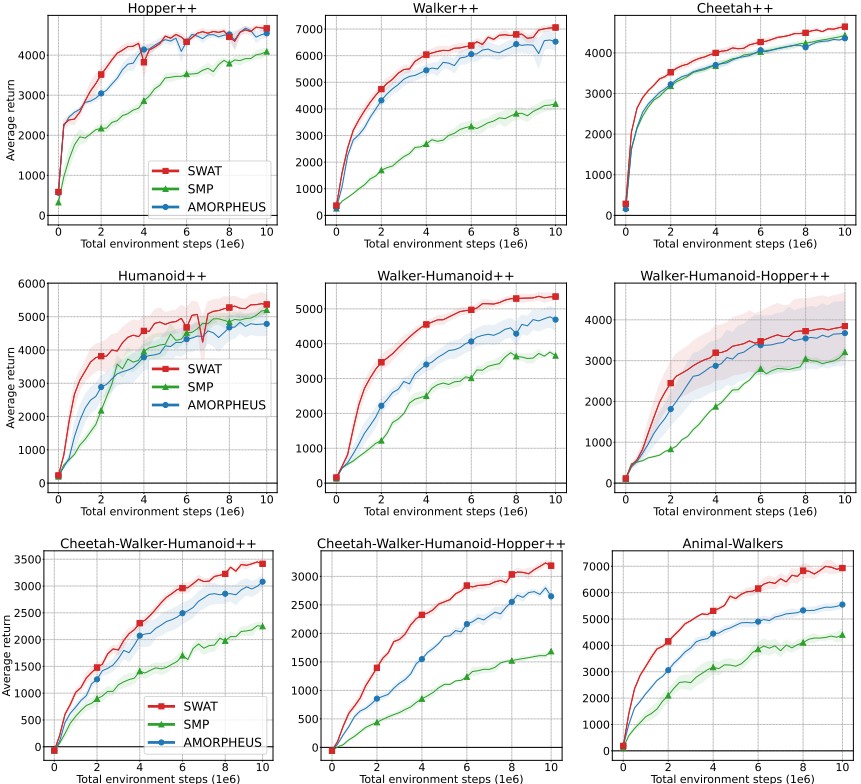

Figure 4: Training curves on 9 environments. We evaluate on 5 different seeds and plot the mean of average returns over all morphologies. The shaded area represents the standard error.

(3) `Cheetah-Walker-Humanoid++`, (4) `Cheetah-Walker-Humanoid-Hopper++` (`CWHH++`), and (5) `Animal-Walkers`[1]. See Appendix A.1 for more details.

We compare our method, **SWAT**, with 2 baselines, the GNN-based method **SMP** (Huang et al., 2020) and the morphology-free self-attention method **AMORPHEUS** (Kurin et al., 2021). We use TD3 (Fujimoto et al., 2018) for training the policy over both baselines and ours for fairness. The policy is trained jointly over all morphologies in various environments, and we run every experiment for 5 random seeds to report the mean and the standard error.

## 5.2 MTRL RESULT

Our results are summarized in Figure 4. As Kurin et al. (2021) pointed out, SMP, which exploits the morphological information but suffers the over-smoothing problem, shows the worst performance among all the methods. On the other hand, AMORPHEUS that discards the morphological information shows better performance than SMP owing to direct communication via self-attention mechanism.

SWAT clearly outperforms the previous state-of-the-art, AMORPHEUS, especially in the cross-domain environments, where different types of morphologies are mixed. Furthermore, the performance gap between SWAT and AMORPHEUS is notably larger in `Animal-Walkers`, which has the greatest morphological diversity. Another noteworthy observation is that SWAT achieves the best mean performance as well as converges faster than other baselines in both in-domain and cross-domain environments. These results consistently demonstrate that the effectiveness of our embedding method increases as more various types of robots are jointly trained. We speculate the reason for the effectiveness is that the PE and RE encourage our model to transfer commonalities

---

[1]Unlike other environments whose variants are in fact subsets of the intact morphology, `Animal-Walkers` consists of fundamentally different morphologies, such as a Walker, Horse, and Ostrich.

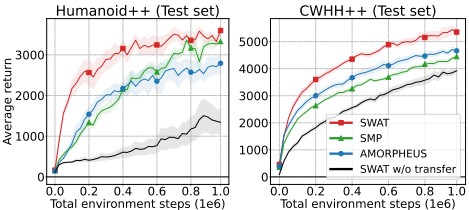 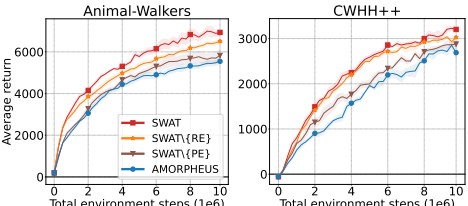

Figure 5: Performance in transfer learning.   Figure 6: Ablation study on PE and RE.

among resemblant partial morphologies (e.g., the gait patterns of the lower bodies of the Walker and Humanoid), while discriminating against unrelated ones (the legs of Hopper from those of Walker). By utilizing these structural embeddings across all environments, our model can learn faster and achieve higher final returns than those without the structural embeddings.

### 5.3 TRANSFER LEARNING RESULT

In this section, we benchmark SWAT in a transfer learning setting where the policy is trained in various tasks and then transferred to another downstream task, which is a common learning strategy in CV and NLP. We compare SWAT with SMP, AMORPHEUS, and **SWAT w/o transfer**, i.e., SWAT without pretraining. All models except for SWAT w/o transfer are pretrained in the train environments and transferred to the test environment with morphology unseen to them. We evaluate them on `Humanoid++`, which has the largest number of limbs among the in-domain, and on `CWHH++`, which contains all in-domain environments. Note that we only train them for 1 million time steps during transfer learning.

As shown in Figure 5, with all pretrained models outperforming SWAT w/o transfer, SWAT learns considerably faster than other baselines, showing much higher sample efficiency, and again outperforms them in average returns. Similar to multi-task learning settings, the useful knowledge obtained from the training tasks can be effectively transferred to the unseen tasks by means of the positional and relational information, enabling quick adaptation with fewer samples. Although our main focus is MTRL and transfer learning, we also conduct experiments on zero-shot setting in Appendix A.4.

### 5.4 ABLATION STUDY

In this section, we conduct additional experiments to analyze how the positional and relational information affects the performance of our model respectively. We ablate the PE and the RE, and both of them from SWAT, rendering our model identical to AMORPHEUS. We evaluate these on `Animal-Walkers` and `CWHH++`, where we expect for their morphological diversity to incite each embedding to play a more crucial role than in other environments.

As shown in Figure 6, the performance of SWAT degrades when the embeddings are removed one by one. This result supports that our proposed embeddings indeed play important roles in different manners. In other words, we need both positional and relational information to take full advantage of the given morphology. Meanwhile, as it can be seen from SWAT\\{RE}, i.e., SWAT with the positional embedding only, versus SWAT\\{PE}, i.e., SWAT with the relational embedding only, the performance gains from the positional information are greater than ones from relational information. We conjecture that the self-attention, originally devised to learn the implicit relations among nodes, might capture some relational information but cannot the absolute positional information. Further ablation studies about each component of PE and RE are provided in Appendix A.5.

### 5.5 BEHAVIOR ANALYSIS

In this section, we investigate how the agent that is jointly trained in the various environments acts in a single environment. To examine this, we visualize trajectories of the agent trained in `CWHH++`. Figure 7(a) compares the mean performance of AMORPHEUS and SWAT in the single environment of intact Humanoid with the largest number of 9 limbs.

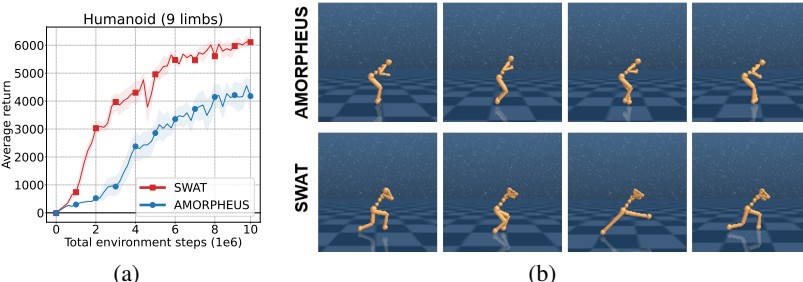

Figure 7: (a) AMORPHEUS and SWAT performance. (b) We visualize trajectories from AMORPHEUS and SWAT on Humanoid with 9 limbs in `CWHH++`.

As shown in Figure 7(b), AMORPHEUS does not learn to walk but instead jumps to move forward like Hopper. This result shows the unexpected effect of multi-task learning. In multi-task learning, the knowledge from other tasks is not always beneficial to the task of interest; therefore, careless exploitation of such knowledge can hamper learning rather than help it. That is, the knowledge obtained in Hopper environments hinders learning to walk in the case of AMORPHEUS. On the other hand, SWAT successfully learns to run since the structural embeddings allow it to transfer the appropriate knowledge exclusively where necessary. We speculate this is possible as the embeddings which have learned similar limbs across different agents are mapped closely to one another. We visualize the structural embeddings in Appendix A.3.

## 6 RELATED WORK

There have been lots of studies about MTRL with the homogeneous setting where the state and action spaces are the same across tasks but each task has a different objective (Rusu et al., 2016; Parisotto et al., 2016; Teh et al., 2017; Kalashnikov et al., 2021). The inhomogeneous setting has emerged in recent years and explored by several works. Devin et al. (2017) decompose the policy model into task-specific modules and morphology-specific ones, then reuse the combination of them for various settings. D'Eramo et al. (2020) use a shared module over all tasks that is combined with a task-specific encoder and decoder to learn a policy per task. Those approaches based on the separated module have a limitation in that they require a larger number of model parameters as the number of tasks increases. Chen et al. (2018) encodes agent morphology in a feature vector and learns the policy for different morphologies conditioned on the state and feature vector.

Another line of work that is closely related to ours is the modular approach. Wang et al. (2018) and Pathak et al. (2019) represent the agent morphology as a graph and utilize GNNs as the policy network in order to tackle the inhomogeneous setting. Both of their studies show that the GNN-based policy has large benefits over a monolithic policy in inhomogeneous MTRL. Recently, Huang et al. (2020) present SMP based on GNNs for the inhomogeneous setting. They assume each agent as a tree and propose a both-way MP scheme where the messages are propagated from leaves to root (bottom-up) and from root to leaves (top-down). Kurin et al. (2021) pose a concern about the MP in GNNs due to the over-smoothing problem, and propose AMORPHEUS, which adopts self-attention mechanism rather than GNNs for direct communication in exchange for leveraging the morphological information. In contrast, we utilize the morphological information through the structural embeddings, allowing direct communication while taking advantage of the structural bias.

## 7 CONCLUSION

In this paper, we argue that the knowledge of agent morphology plays an important role in modular MTRL and present SWAT that leverages structural biases arising from the morphology. Instead of the MP scheme, we encode the structural information through the medium of the traversal-based PE and the graph-based RE, which can be easily incorporated into the transformer-based policy. We empirically demonstrate that our method achieves the state-of-the-art performance in inhomogeneous robot locomotion MTRL benchmarks as well as in transfer learning settings.

ACKNOWLEDGMENTS

The authors thank Hyolim Kang for contributing to the improvement of SWAT. This work was supported by the National Research Foundation (NRF) of Korea (NRF-2019R1A2C1087634, NRF-2021M3I1A1097938) and Institute of Information & communications Technology Planning & Evaluation (IITP) grant funded by the Korea government (MSIT) (No.2019-0-00075, No.2020-0-00940, No.2021-0-02068)

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

# A  APPENDIX

## A.1  ENVIRONMENT DESCRIPTION

Following Kurin et al. (2021), we conduct experiments on benchmarks as in Table 2. Huang et al. (2020) proposed `Hopper++`, `Walker++`, `Cheetah++`, and `Humanoid++`, removing some parts from the intact morphology, while every morphology is able to hop, walk, or run, i.e., they do not contain the morphology that is unable to go forward such as the robot with only arms or the one lacking its torso.

To explain the name of environment in more detail, the first part means its intact morphology and the middle part means the number of limbs belongs to its morphology, and the last part means the missing limbs. For example, `humanoid_2d_7_right_leg` originates from Humanoid and consists of 7 limbs, lacking right legs, both right thigh and right shin. Note that we dub `Walker` in Wang et al. (2018) as `Animal-Walkers`, because it is quite confusing in that `Walker++` also contains the variants of Walker. Since `Animal-Walkers` consists of various animal-shaped morphology, we distinguish it from `Walker`.

All the environments provides the morphology of the robot as a graph-structured data that is extracted from the corresponding MuJoCo XML file. The reward is given by the distance the agent move forward, penalized by the norm of action, i.e., the sum of torque values in all actuators. And it terminates when the agent slips and cannot proceed anymore or it succeeds to survive 1,000 time steps.

## A.2  IMPLEMENTATION DETAIL

We implement SWAT based on AMORPHEUS which is built on *TransformerEncoder* from Py-Torch, sharing the codebase of SMP. Additionally, we simply modify *TransformerEncoder* to incorporate with PE and RE, enabling relational embedding to be added per head. We use 3 independent embedding layers for each traversal to learn PE, where each embedding size has the equal size, i.e., summation of all PEs per traversal is equal to the embedding size of final PE. And we use a simple fully-connected layer for $g_\phi$ in Equation 4 to learn the graph-based RE. SWAT shares hyperparameter setting used in Kurin et al. (2021) as shown in Table 1, except the replay buffer size due to the memory efficiency. Also, we use the residual connection as reported in Kurin et al. (2021).

| Hyperparemeter | Value |
|---|---|
| Learning rate | 0.0001 |
| Gradient clipping | 0.1 |
| Normalization | LayerNorm |
| Attention layers | 3 |
| Attention heads | 2 |
| Attention hidden size | 256 |
| Encoder output size | 128 |
| Mini-batch size | 100 |
| Replay buffer size | 500K |
| Embedding size | 128 |

Table 1: Hyperparameter setting in SWAT

| Environment | Train set | Test set |
|---|---|---|
| `Hopper++` | | |
| | `hopper_3` | |
| | `hopper_4` | |
| | `hopper_5` | |
| `Walker++` | | |
| | `walker_2_main` | `walker_3_main` |
| | `walker_4_main` | `walker_6_main` |
| | `walker_5_main` | |
| | `walker_7_main` | |
| `Cheetah++` | | |
| | `cheetah_2_back` | `cheetah_3_balanced` |
| | `cheetah_2_front` | `cheetah_5_back` |
| | `cheetah_3_back` | `cheetah_6_front` |
| | `cheetah_3_front` | |
| | `cheetah_4_allback` | |
| | `cheetah_4_allfront` | |
| | `cheetah_4_back` | |
| | `cheetah_4_front` | |
| | `cheetah_5_balanced` | |
| | `cheetah_5_front` | |
| | `cheetah_6_back` | |
| | `cheetah_7_full` | |
| `Humanoid++` | | |
| | `humanoid_2d_7_left_arm` | `humanoid_2d_7_left_leg` |
| | `humanoid_2d_7_lower_arms` | `humanoid_2d_8_right_knee` |
| | `humanoid_2d_7_right_arm` | |
| | `humanoid_2d_7_right_leg` | |
| | `humanoid_2d_8_left_knee` | |
| | `humanoid_2d_9_full` | |
| `Walker-Humanoid++ (WH++)` | | |
| | Union of `Walker++` and `Humanoid++`. | |
| `Walker-Humanoid-Hopper++ (WHH++)` | | |
| | Union of `Walker++`, `Humanoid++`, and `Hopper++`. | |
| `Cheetah-Walker-Humanoid++ (CWH++)` | | |
| | Union of `Cheetah++`, `Walker++`, and `Humanoid++`. | |
| `Cheetah-Walker-Humanoid-Hopper++ (CWHH++)` | | |
| | Union of `Cheetah++`, `Walker++`, `Humanoid++`, and `Hopper++`. | |
| `Animal-Walkers` | | |
| | `FullCheetah (Wolf)` | |
| | `Hopper` | |
| | `HalfCheetah (Horse)` | |
| | `HalfHumanoid` | |
| | `Ostrich` | |

Table 2: Full list of environments.

## A.3 STRUCTURAL EMBEDDING

We visualize both PE and RE in SWAT learned in cross-domain environments, `CWHH++`. Figure 8 shows that the PEs of limbs placed at similar position in morphology are mapped closely. Figure 9, 10, and 11 shows SWAT learns relational information among nodes through $\mathbf{R}_{lap}$, $\mathbf{R}_{spd}$ and $\mathbf{R}_{ppr}$, which considers not only local but global information of graph.

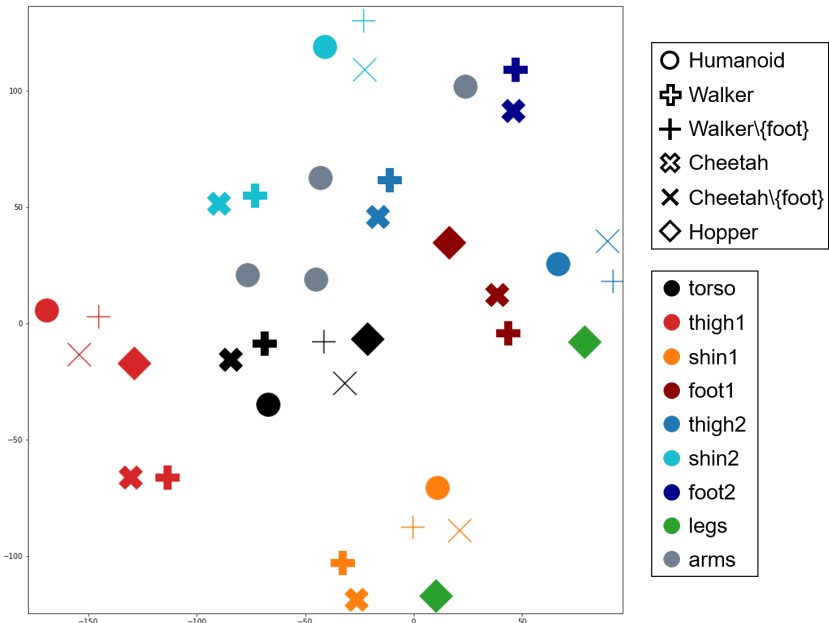

Figure 8: t-SNE visualization of positional embeddings learned from multiple robots with various morphology. The points are corresponded to the reduced positional embedding of each node in different morphologies. The shape denotes the morphology it belongs to and the color denotes its limb type. Walker(Cheetah)\{foot} are the variants lacking of feet.

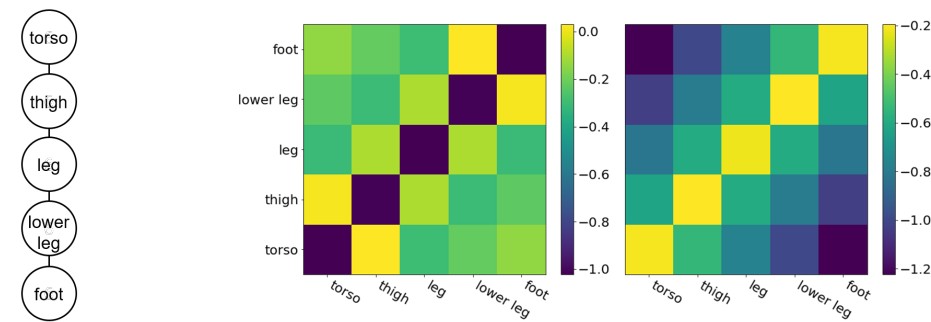

Figure 9: Illustration of the relational embeddings from two heads on Hopper (5 limbs)

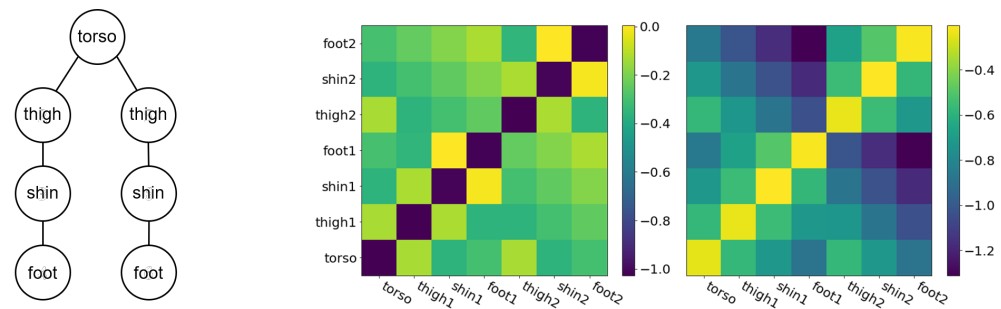

Figure 10: Illustration of the relational embeddings from two heads on Walker (7 limbs)

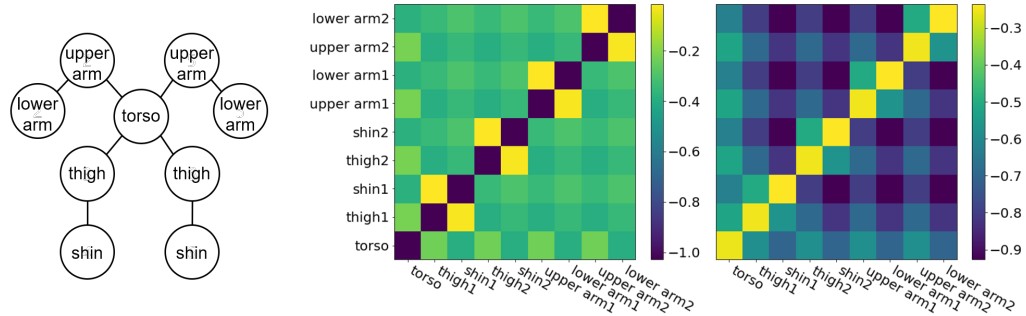

Figure 11: Illustration of the relational embeddings from two heads on Humanoid (9 limbs)

## A.4   ZERO-SHOT EVALUATION

Table 3 and 4 show zero-shot evaluation results on in-domain setting and cross-domain setting respectively. For in-domain setting, each policy is trained on `Walker++`, `Humanoid++`, `Cheetah++` and evaluated on their test set respectively. For example, SWAT trained on the train set of `Walker++` is tested on `walker_3_main` and `walker_6_main`. Note that we omit the lacking part in the name of morphologies in the Table 3 and 4. For cross-domain setting, each policy is trained on `CWHH++` and evaluated on its test set, i.e., the union of test sets of `Walker++`, `Humanoid++`, `Cheetah++`. We evaluate by the average performance and the standard error over 3 seeds, where each seed is evaluated on 100 rollouts. As in Kurin et al. (2021), the high variance due to instability in generalization can be observed in SWAT as well as in SMP and AMORPHEUS. Therefore, it would be hard to say which one is statistically advantageous to the others in zero-shot generalization task.

| | SWAT | AMORPHEUS | SMP |
|---|---|---|---|
| walker_3 | $220.49 \pm 66.62$ | $264.04 \pm 37.39$ | $218.07 \pm 49.27$ |
| walker_6 | $396.10 \pm 136.98$ | $745.28 \pm 635.61$ | $900.64 \pm 172.95$ |
| humanoid_2d_7 | $2428.38 \pm 1252.61$ | $659.54 \pm 519.64$ | $3042.23 \pm 1154.78$ |
| humanoid_2d_8 | $4117.4 \pm 171.54$ | $3624.95 \pm 1035.22$ | $3798.49 \pm 137.23$ |
| cheetah_3 | $10.94 \pm 224.54$ | $171.07 \pm 230.04$ | $139.09 \pm 268.49$ |
| cheetah_5 | $1011.95 \pm 945.82$ | $3027.78 \pm 735.40$ | $2768.24 \pm 229.80$ |
| cheetah_6 | $2610.97 \pm 747.48$ | $4989.3 \pm 257.68$ | $5861.86 \pm 307.39$ |

Table 3: Comparison in zero-shot evaluation on in-domain test set.

| | SWAT | AMORPHEUS | SMP |
|---|---|---|---|
| walker_3 | $256.62 \pm 133.57$ | $216.73 \pm 40.24$ | $52.30 \pm 24.39$ |
| walker_6 | $1564.45 \pm 1229.78$ | $3209.67 \pm 1425.86$ | $852.41 \pm 363.50$ |
| humanoid_2d_7 | $1916.45 \pm 1358.3$ | $2590.07 \pm 905.40$ | $532.02 \pm 314.94$ |
| humanoid_2d_8 | $4040.86 \pm 296.85$ | $1201.67 \pm 565.64$ | $1205.34 \pm 683.62$ |
| cheetah_3 | $-89.63 \pm 41.44$ | $50.13 \pm 199.91$ | $133.56 \pm 68.88$ |
| cheetah_5 | $855.30 \pm 243.15$ | $1842.01 \pm 40.70$ | $290.19 \pm 52.70$ |
| cheetah_6 | $2625.68 \pm 291.62$ | $3113.09 \pm 262.48$ | $2778.24 \pm 391.44$ |

Table 4: Comparison in zero-shot evaluation on cross-domain test set.

## A.5 FURTHER ABLATION STUDY ON EMBEDDING COMPONENTS

In this section, we further conduct experiments to analyze how each component in the positional and relational embedding affects the performance of our model. We firstly verify the usefulness of components in the PE, which are pre-order (PRE), in-order (IN) and post-order (POST) traversal. As shown in Figure 12, SWAT\{POST,IN} shows the worst performance due to ambiguity, i.e., a binary tree cannot be reconstructed only with pre-order traversal. Using in-order traversal along with pre-order traversal is theoretically sufficient to recover a binary tree, so giving an additional post-order seems to be redundant. However, as it can be seen from SWAT versus SWAT\{POST}, the empirical result shows that the additional information actually helps learning more efficiently in some cases. This is because although redundant, they can provide positional information in different perspectives, making the model easier to figure out node positions. We thus decided to use all three traversals in the main experiments. Similar to the PE, we also check the effect of each component in the RE, which are PPR, SPD and LAP, shown in Figure 13. While the performance gaps among 3 variants are not large, SWAT consistently outperformed others. We thus used all three features for RE in the main experiments although they provide overlapping information.

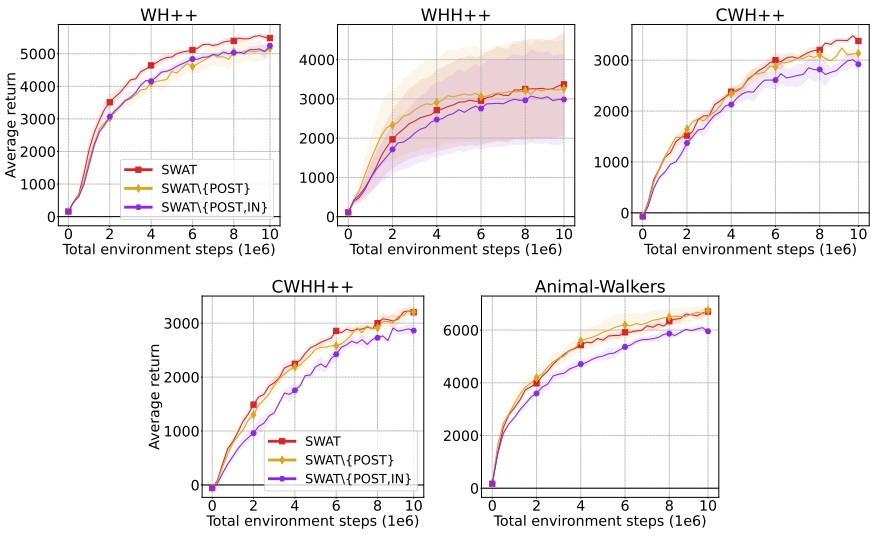

Figure 12: Ablation study on traversal-based positional embedding.

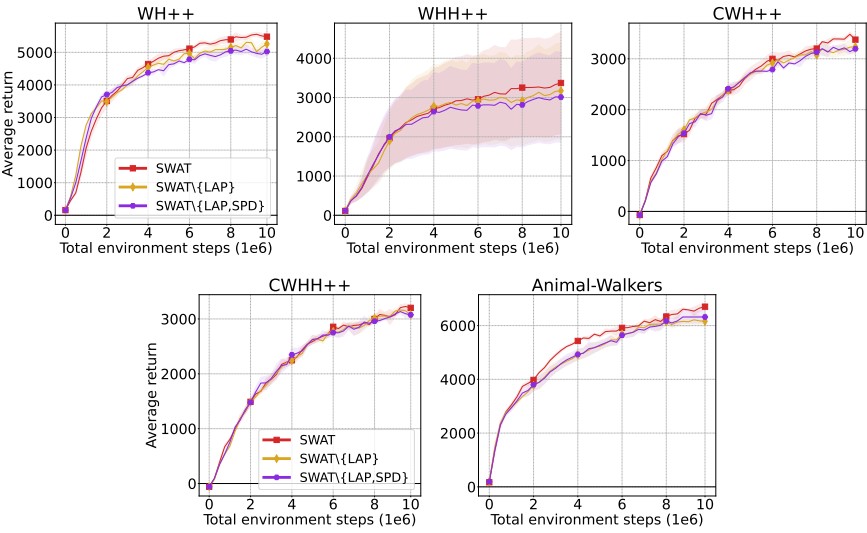

Figure 13: Ablation study on graph-based relational embedding.

