# OpenReview forum: "Structure-Aware Transformer Policy for Inhomogeneous Multi-Task Reinforcement Learning"
_ICLR.cc/2022/Conference — ICLR 2022 Poster_

### Official Review · Reviewer_BRC4 · 2021-10-31

**Correctness:** 4
**Technical Novelty And Significance:** 3
**Empirical Novelty And Significance:** 3
**Recommendation:** 6
**Confidence:** 2

**Main Review:**

# Strengths

- Significant performance increase from previous state-of-the-art in multi-task scenarios with diverse morphologies.
- A relevant perspective shift from SOTA in inhomogenous MTRL: an appropriate structural encoding can improve performance than capturing no structural information in the policy.
- Well-written with the relevant related work.
- Qualitative analysis of what the structural embeddings capture.

# Weakness

- While the paper shows ablation experiments by excluding PE and RE from SWAT, it does not discuss the effect of different tree-traversals in PE and graph connectivity representations in RE. It seems that there is redundant information in the proposed PE and RE representations. For example, is there any extra information provided by Normalized Laplacian and SPD that PPR cannot capture? What if only PPR is used to represent the graph-based RE? Further ablation analysis would be required to answer if  embedding that include different and overlapping information are necessary for generalization and transfer.
- Is SWAT is computationally more expensive than the baseline approaches? SWAT requires computing additional PE and RE embeddings compared to AMORPHEUS and uses of quadratic attention compared to SMP. What would be the wall-clock time for taking an "environment step" in each of the three models? What is the overhead computation required for ε-discounted random walk for PPR embedding? A discussion in this regard would be helpful for reproducibility and future research directions.
- [Clarification needed] In Transfer learning, the train set of environments is a bit unclear for Humanoid++ and CWHH++ test tasks. Is it consisting of all the 9 train environments except the test environment as shown in Appendix A.2 Table 2? Has the policy for test setting of CWHH++ seen the morphologies of Cheetah/Walker/Humanoid/Hopper in the in-domain tasks? Is only difference in train and test setting is an unseen state and action space size?

**Summary Of The Paper:**

The paper proposes SWAT to incorporate the agent's morphology into transformer-based policy in multi-task RL. Assuming that the agent's limbs are nodes and joints are edges of a graph, two embeddings capture the morphological structure: first, positional embedding (PE) to represent the absolute position of limb nodes by tree-traversal (pre-order, in-order, post-order), and second, relational embedding (RE) to represent the relative distances of limb nodes in terms of their graph connectivity (normalized Laplacian, shortest path distance, personalized page rank).

The hypothesis is SWAT's traversal-based PE and graph-based RE enable effective multi-task policy learning in diverse morphological tasks. In its support, the paper demonstrates that SWAT achieves a higher average return than the baselines GNN-based method (SMP) and morphology-free transformer (AMORPHEUS) in Gym Mujoco locomotion tasks. While SWAT and AMORPHEUS are close in performance on four in-domain locomotion tasks, SWAT shows a significantly higher average return than all baselines in five cross-domain tasks with diverse morphologies.
Two ablation experiments analyze the performance impact on excluding PE and RE. The paper also qualitatively examines the behavior patterns for SWAT and AMORPHEUS policies trained in cross-domain (CWHH++) and tested in a single task (Humanoid).

**Summary Of The Review:**

I enjoyed the proposed simple additions to existing state-of-the-art, showing how certain body morphology encodings can significantly improve the performance. While this work again shows that structural information can be helpful in MTRL, it brings up the question of the appropriate ways to encode the structure for inhomogeneous MTRL for successful policy learning.

While there seems no proven "optimal" way for encoding structural information, the proposed PE and RE features in transformer-based policy seem an excellent first step.

---

> ### Author Response · Authors · 2021-11-20
> **Response to Reviewer BRC4**
>
> Thank you for your constructive feedback.
>
> **[Technical novelty]**
> The novelty lies in (1) proposing a novel embedding scheme that utilizes tree-traversal order to represent the node positional information of the graph-structured data and (2) use it for modular RL, which have not appeared in the literature to the best of our knowledge.
>
> **[Discussion for the effect of different tree-traversals in PE and graph connectivity representations in RE]**
> We further conducted ablation experiments to analyze how each component in the positional and relational embedding affects the performance of our model in Appendix A.5.
>
> **[Difference in computational costs from baselines and our method]**
> In practice, there was no significant difference in wall-clock computation time, between self-attention approach (SWAT and AMORPHEUS) and GNN approach (SMP). Although a quadratic complexity of self-attention is the main bottleneck of transformers with long sentences in NLP, it was not a big problem in our inhomogenous control tasks since the number of nodes was maximum 13. In addition, this computation is parallelized in modern implementations using GPUs. In fact, the wall-clock time per environment step of SWAT, AMORPHEUS, and SMP in the CWHH++ were 44.5 ms, 43.7 ms, 46.4 ms respectively.
> SWAT requires extracting features for embeddings such as PPR, which can be done in $O(nk)$ where $n$ is the number of nodes and $k$ is the number of iterations, but they are pre-computed before the actual training of the policy. Thus the computational overhead is negligible during the RL loop.
> We will publicly share our code for reproducibility when our work no longer has to remain anonymous.
>
> **[Clarification for train and test set of the benchmark]**
> All the environments in the standard benchmark suite have their own train / test tasksets as shown in Table 2. For example, Humanoid++ environment is composed of *humanoid_2d_7_left_arm*, *humanoid_2d_7_lower_arms*, *humanoid_2d_7_right_arm*, *humanoid_2d_7_right_leg*, *humanoid_2d_8_left_knee*, *humanoid_2d_9_full* for training taskset, and *humanoid_2d_7_left_leg*, *humanoid_2d_8_right_knee* for test taskset. CWHH++ environment is composed of training tasksets from Hopper++, Walker++, Cheetah++, and Humanoid++ for training taskset (i.e., union), and test tasksets from the same 4 environments for test taskset. So the agent has seen the morphology during training for the in-domain and cross-domain transfer learning experiments, which was the evaluation protocol in [1].
>
> [1] Kurin et. al., My Body Is A Cage: The Role of Morphology in Graph-based Incompatible Control, ICLR, 2021.

---

> > ### Comment · Reviewer_BRC4 · 2021-11-29
> > **Thanks for the rebuttal response**
> >
> > I thank the authors for addressing the concerns and adding the ablation experiment to analyze the design choices for PE and RE embeddings. I have increased my score for technical novelty as the proposed approach shows that adding graph structure information is better than previous methods, which were only transformer-based and only GNN-based policies.

---

### Official Review · Reviewer_bVEF · 2021-11-01

**Correctness:** 3
**Technical Novelty And Significance:** 3
**Empirical Novelty And Significance:** 3
**Recommendation:** 6
**Confidence:** 4

**Main Review:**

**Strength:**

- The paper is well-motivated in that structural and morphological information about the robot could help improve the performance in MTRL and transfer learning, as this information can make the transformer-based policy more specialized for each morphology.
- The proposed traversal-based embedding seems new and achieves good performance as shown in the ablation studies. The overall method seems quite easy to implement and has the potential to improve other GNN-based tasks.
- The proposed method outperforms the SOTA in both MTRL and transfer learning.
- The paper is well-written and easy to follow. The figures (e.g., Fig. 3) are nicely done and convey the main idea clearly.

**Weakness:**

- Base on its design, I feel the proposed traversal-based positional embedding is not very suitable for generalization. The index of each joint is quite global and not actually transferrable among different morphologies. For example, a foot joint with index i in robot A may have index j in robot B, but index j may represent a completely different joint (e.g., head) in robot A, so the learned embedding p_j (learned for the head joint) may not be suitable for the foot joint in robot B. I think the proposed method can still achieve good performance in MTRL because the training in MTRL allows the policy to overfit to the learnable positional embeddings. But the proposed embedding has limited generalization ability so it is not suitable for tasks like zero-shot policy transfer which has been demonstrated in [Huang et al., 2020]. I think some experiments on zero-shot policy transfer are required to shed some light on this, or the paper should discuss the limitation in generalization.
- The paper spent a fair amount of space on relational embedding (RE), however, the performance gain from it seems a bit small. Most performance gain seems to come from the positional embedding (PE). As discussed in my first point, is it because the PEs already allows the network to overfit to them in MTRL?
- Ablation studies on all the environments should be provided, especially because the proposed RE seems not very effective in Fig. 6. Additionally, both the PE and RE have multiple components. It would be nice to do some ablation studies in those components to understand their usefulness.

**Comments:**

Typo:

- "the performance gains from the positional information(SWAT\RE) are greater than ones from relational information (SWAT\PE)" —-  switch RE and PE?

**Summary Of The Paper:**

This paper proposes investigates the role of positional and relational embeddings in transformer-based policies for multi-task reinforcement learning (MTRL) across different morphologies. Unlike prior works that completely discard the structural information of the robot in transformer-based policy, this paper proposes to leverage structural information using traversal-based positional embedding and graph-based relational embeddings. Experiments are performed for MTRL as well as transfer learning on several gym environments, which demonstrate the effectiveness of the proposed method against the baselines.

**Summary Of The Review:**

Overall, this is a well-motivated paper with good performance. The proposed traversal-based PE seems quite useful for MTRL task. However, the proposed RE seems less useful and the ablation studies are not quite sufficient. I also have doubts in terms of the PE's generalization ability, which could be addressed through zero-shot policy transfer experiments. I vote for a weak accept for now, but may change my score based on the authors' response.

---

> ### Author Response · Authors · 2021-11-20
> **Response to Reviewer bVEF**
>
> Thank you for your constructive feedback.
>
> **[Generalization issue & Zero-shot evaluation]**
> We added generalization experiments using zero-shot transfer setting in Appendix A.4. SWAT does not show improved results on the zero-shot setting, compared to the multi-task and transfer learning setting. It may be due to the reason you pointed out, i.e., some index may represent totally different joint, and therefore it cannot generalize well to an unseen robot without additional training. However, as [1] pointed out, all the previous works also suffered from instability issue (high variance) in zero-shot generalization. Extending to a more general embedding scheme that can overcome the instability issue would be an interesting direction for future work, e.g. augmentation strategy for the training taskset to improve zero-shot transfer performance.
>
> **[The reason why performance gain from PE is larger than RE]**
> As you mentioned, one of the reasons can be the overfitting. Additionally, depending on tasks, the relational embedding can have less effect on performance than the positional embedding as shown in [2]. We conjecture that the self-attention itself, originally devised to learn the implicit relations among nodes, captures some of the relational information that we explicitly give through RE. On the other hand, the absolute positional information cannot be captured through self-attention unless it is given explicitly by PE.
>
> **[Discussion on the effect of different tree-traversals in PE and graph connectivity representations in RE]**
> We further conduct experiments to analyze how each component in the positional and relational embedding affects the performance of our model in Appendix A.5. As shown in Figure 4 in section 5.2, the performance gap between AMORPHEUS and SWAT was not large in the in-domain environments (Hopper++, Walker++, Cheetah++, and Humanoid++) so we conducted ablation studies on all the cross-domain environments (WH++, WHH++, CWH++, CWHH++, and Animal-Walkers).
>
> **[Minor typo]**
> Actually, we used ‘\’ to mean ‘set difference’. We clarified it in the main text.
>
> [1] Kurin et. al., My Body Is A Cage: The Role of Morphology in Graph-based Incompatible Control, ICLR, 2021.
> [2] Wu et al., Rethinking and Improving Relative Position Encoding for Vision Transformer, ICCV, 2021.

---

> > ### Comment · Reviewer_bVEF · 2021-11-30
> > **Thank you for the response**
> >
> > Thank you for the response and additional experiments. I will keep my current score since I feel the paper does provide good empirical performance although there is some concern about the generalizability of this approach for other tasks. I hope the authors could discuss these points in the main paper to let the reader know what to expect.

---

### Official Review · Reviewer_xDqR · 2021-11-02

**Correctness:** 3
**Technical Novelty And Significance:** 2
**Empirical Novelty And Significance:** 3
**Recommendation:** 6
**Confidence:** 4

**Main Review:**

Strength:
- The paper is well written and easy to read.
- The proposed approach is straightforward.
- The experiments show that the proposed approach outperformed competitive baselines such as SMP and AMORPHEUS.

Weakness:
- The reviewer has some concerns about the novelty of the paper. Positional encoding for transformers has been widely used and recognized in NLP and computer vision literature. The paper’s main contribution is showing that transformers with positional encoding outperforms the ones without the positional encoding in MTRL tasks. The reviewer thinks it is reasonable but the contribution seems somewhat incremental.

- The reviewer has some questions about the technical contents. Please see the comments below:
   - To reconstruct a binary tree, inorder traversal along with preorder or postorder traversal is sufficient. However, in Eq (3) all the three traversal results are aggregated together. What is the purpose of adding redundant information?
   - In inhomogeneous MTRL, each task has different state dimensions. However, in Section 4.3, it says a single MLP layer is shared across all nodes. Is some padding mechanism used here to make a single MLP layer applicable to all nodes?

- Experiments
   - For relational encoding, the authors combine three features: (1) normalized graph Laplacian (2) shortest path distance, (3) page rank. It would be great to check the effectiveness of each of the features vai an ablation study.

  - In Section A2, the authors mentioned that they used a different replay buffer size from the baseline. For a fair comparison, why not using the same size?


**Summary Of The Paper:**

This paper addresses the inhomogeneous multi-task reinforcement learning (MTRL) problem. Following existing works (Huang et al. 2020, Kurin et al. 2021), a policy is represented as a graph. Each node in the graph is an identical modular neural network. The current state-of-the-art approach,  (Kurin et al. 2021),  uses a self-attention mechanism which allows direct communication between nodes. The author argued that the sel-attention mechanism discards the morphological information, which may be critical for agent learning and better perfromance. Therefore, they proposed to encode the morphological information via traversal-based positional embedding and graph-based relational embedding. The proposed approach is simple yet outperforms the state-of-the-art approaches on the MUJOCO MTRL environments.


**Summary Of The Review:**

The reviewer thinks this paper is well written and the proposed positional encoding is simple yet seems effective. However, the reviewer has some concerns about the novelty of the paper, since positional encoding is widely used along with transformers.

---

> ### Author Response · Authors · 2021-11-20
> **Response to Reviewer xDqR**
>
> Thank you for your constructive feedback.
>
> **[Novelty of positional encoding in transformer]**
> As you pointed out, the concept of positional encoding is widely used in transformer for NLP and CV. However, it is still not straightforward to represent the graph-structured data with standard positional embedding used in language models. Our main contributions lies in (1) proposing a novel embedding scheme that utilizes tree-traversal order to represent the node positional information of the graph-structured data and (2) use it for modular RL, which have not appeared in the literature to the best of our knowledge.
>
> **[Purpose of using additional traversal more than 2 traversals]**
> We conduct the ablation study on the positional embedding in Appendix A.5. Using in-order traversal along with pre-order traversal is theoretically sufficient to recover a binary tree, so giving an additional post-order seems to be redundant. However, as it can be seen from SWAT versus SWAT without post-order traversal in Appendix A.5, the empirical result shows that the additional information actually helps learning more efficiently in some cases.  This is because although redundant, they can provide positional information in different perspectives, making the model easier to figure out node positions. We thus decided to use all three traversals in the main experiments.
>
> **[How to handle the different state-action dimension with a single MLP module?]**
> In inhomogeneous MTRL, the state-action dimensions are different across the tasks, since each agent consists of different numbers of nodes. However, the dimension of a local state (the state of each node) in the benchmark we used is equal across the tasks, so a single MLP module can be shared across every node for inhomogeneous MTRL without the padding mechanism. If the local state for each node has different dimensions, we can utilize some padding mechanism.
> We will publicly share our code for reproducibility when our work no longer has to remain anonymous.
>
> **[Ablation study on 3 features for relational embedding]**
> We further conducted experiments to analyze how each component in the positional and relational embedding affects the performance of our model in Appendix A.5.
>
> **[Different size of replay buffer from baselines]**
> We used the same size of replay buffer across the baselines and our method in all our experiments for a fair comparison. We decreased the size of the replay buffer from the original size used in prior works, because the original size requires too much memory for our computing infrastructure. Although we decreased it, we observed that it does not significantly affect the performance of all baselines compared to the result in [1].
>
> [1] Kurin et. al., My Body Is A Cage: The Role of Morphology in Graph-based Incompatible Control, ICLR, 2021.

---

> > ### Comment · Reviewer_xDqR · 2021-11-29
> > **Thanks for the response.**
> >
> > The author response addressed most of my questions. I raised the score to 6.

---

### Official Review · Reviewer_4eRD · 2021-11-03

**Correctness:** 3
**Technical Novelty And Significance:** 2
**Empirical Novelty And Significance:** 2
**Recommendation:** 5
**Confidence:** 3

**Main Review:**

Strengths:

s1) This work introduces traversal-based positional embedding and graph-based relational embedding to encode the morphological information.

s2) The performance of the algorithm significantly outperforms the prior state-of-the-art methods on some of the tasks.

s3) The paper is well-organized and easy to follow.

Weaknesses:

The major weakness of this paper is the experimental evaluation. For figure 4, only three different seeds are used. I would recommend 10 seeds.

**Summary Of The Paper:**

This work introduces a structure-aware transformer for inhomogeneous multi-task reinforcement learning tasks. This work proposes to use traversal-based positional embedding and graph-based relational embedding to encode morphological information. The papers show that their proposed approach outperforms prior state-of-the-art methods on the module multi-task RL benchmarks and transfer learning settings.

**Summary Of The Review:**

This work proposes to use traversal-based positional embedding and graph-based relational embedding to encode morphological information for inhomogeneous multi-task reinforcement learning tasks. The proposed approach seems reasonable. However, the technical novelty is limited. Experiments with more seeds are needed to strengthen the paper.

---

> ### Author Response · Authors · 2021-11-20
> **Response to Reviewer 4eRD**
>
> Thank you for your constructive feedback.
>
> **[Technical novelty]**
> The novelty lies in (1) proposing a novel embedding scheme that utilizes tree-traversal order to represent the node positional information of the graph-structured data and (2) use it for modular RL, which have not appeared in the literature to the best of our knowledge.
>
> **[Number of random seeds]**
>
> Similarly to SMP [1] (4 seeds) and AMORPHEUS [2] (3 seeds), we used 3 random seeds for all experiments.
>
> Since the multi-task learning setting requires rollouts from the various environments, the experiments require a vast amount of computing resources and time (approximately over a week for a server with RTX2080Ti) for each seed. So it was intractable for us to run all the experiments with additional seed within the rebuttal period, in addition to ablation study requests by the reviewers.
>
> However, we will add results from additional seeds in the final version of the paper. We do not expect that the results will differ significantly from those currently reported in the paper.
>
> [1] Huang et. al., One Policy to Control Them All: Shared Modular Policies for Agent-Agnostic Control , ICML, 2020.
> [2] Kurin et. al., My Body Is A Cage: The Role of Morphology in Graph-based Incompatible Control, ICLR, 2021.

---

### Author Response · Authors · 2021-11-20
**Summary of the updates in the revision**

We appreciate the reviewers’ detailed and thoughtful feedbacks. We are excited to further improve SOTA results on modular MTRL via novel embedding methods that reflect the morphology of the agent.

Based on the feedbacks, we have revised our draft by making the following changes:
- We have added a bit more explanation for clarifying the difference between SWAT\\{RE} and SWAT\\{PE} in section 5.4, as suggested by Reviewer bVEF.
- We have added additional zero-shot evaluation in Appendix A.4, as suggested by Reviewer bVEF.
- We have added additional ablation study on the components in positional and relational embedding in Appendix A.5, as suggested by Reviewer BRC4, Reviewer bVEF, and Reviewer xDqR.

If you have any additional questions or concerns to our response, we are more than happy to provide additional responses during the rebuttal period.

---

### Decision · Program_Chairs · 2022-01-20

**Decision:**

Accept (Poster)

**Comment:**

This paper studies the role of positional and relational embedding s for multi-task reinforcement learning with transformer-based policies, The paper is well-motivated, the experiment shows its effectiveness against other competitive methods. In the rebuttal period, the authors solved most of the reviews’ questions such as novelty and ablation studies. There are still some concerns about the generalizability of this approach for other tasks and more experiments are needed.